# Ozone ultrafine bubble water sterilizes *Porphyromonas gingivalis* and neutralizes its virulence factors

Mana Endo[1,2], Hisanori Domon[1,3], Satoru Hirayama[1], Fumio Takizawa[1,2], Akiomi Ushida[4], Koichi Tabeta[2], Yutaka Terao[1,3]*

1 Division of Microbiology and Infectious Diseases, Niigata University Graduate School of Medical and Dental Sciences, Niigata, Japan, 2 Division of Periodontology, Niigata University Graduate School of Medical and Dental Sciences, Niigata, Japan, 3 Center for Advanced Oral Science, Niigata University Graduate School of Medical and Dental Sciences, Niigata, Japan, 4 Institute of Science and Technology, Niigata University, Niigata, Japan

* terao@dent.niigata-u.ac.jp

## Abstract

Periodontitis is caused by Gram-negative bacteria. *Porphyromonas gingivalis,* a major Gram-negative periodontal pathogen, produces virulence factors such as gingipains, lipopolysaccharide (LPS), and lipoproteins, which contribute to tissue destruction. Ozone ultrafine bubble water (OUFBW) has been studied for its antimicrobial effects against various bacteria and toxin protein inactivation. This present study aimed to explore the effects of OUFBW on *P. gingivalis* and its virulence factors. OUFBW was generated and applied to *P. gingivalis* to assess bactericidal activity. OUFBW effects on morphological changes in *P. gingivalis* and its membrane vesicles (MVs) were analyzed using transmission electron microscopy. Gingipain activities in OUFBW-treated *P. gingivalis* culture supernatant was tested using fluorogenic substrates and endogenous substrates, E-cadherin and IL-6. Degradation of OUFBW-treated gingipains was analyzed by silver staining and western blotting. Effects of OUFBW on *P. gingivalis* lipoproteins and LPS were evaluated using HEK-Blue cells expressing Toll-like receptor 2 (TLR2) and TLR4, respectively. Cytotoxicity of OUFBW on human gingival cells was analyzed using an MTT assay. OUFBW disrupted the inner membrane of *P. gingivalis*, leading to elimination of the bacterium and reduction of MVs. OUFBW also decreased gingipain activities and inhibited gingipain-induced degradation of E-cadherin and IL-6 due to gingipain breakdown. Additionally, OUFBW suppressed lipoprotein-induced TLR2 activation but had no effect on LPS-mediated TLR4 signaling. OUFBW showed low cytotoxicity in human gingival cells. Our findings indicate that OUFBW can sterilize *P. gingivalis* and inactivate gingipains and lipoproteins. These results suggest that OUFBW would be used to disinfect periodontal treatment instruments.

**Data availability statement:** All relevant data are within the paper and its Supporting Information files.

**Funding:** This work was supported by the Japan Society for the Promotion of Science KAKENHI (JP22K09923, JP23H00445, and JP23K18355) and JST SPRING Grant (No. 161040-J24H0003). The funders had no role in study design, data collection and interpretation, or the decision to submit the work for publication.

**Competing interests:** The authors have declared that no competing interests exist.

## Introduction

Periodontitis is an oral infectious and inflammatory disease caused primarily by Gram-negative bacteria. Approximately 500 bacterial taxa inhabit the subgingival environment and their interaction with the host innate immune system triggers an inflammatory response, leading to periodontal tissue destruction and bone loss [1]. Reportedly, approximately 538 million people had severe periodontitis and 276 million experienced tooth loss globally in 2015 [2]. One of the most widely studied periodontopathogens is *Porphyromonas gingivalis*, a Gram-negative obligate anaerobic bacterium, which subverts the host immune response through various virulence factors such as gingipains, lipopolysaccharide (LPS), and lipoproteins [3].

Gingipains, the major cysteine proteases of *P. gingivalis*, account for 85% of the total extracellular proteolytic activity of the bacterium [4]. There are two types of gingipains: arginine-specific protease (Rgp) and lysine-specific protease (Kgp). Rgps are encoded by the *rgpA* and *rgpB* genes, and Kgp is encoded by the *kgp* gene. These gingipains are located on the surface of the bacterial outer membrane, contained within membrane vesicles (MVs), or released extracellularly as secretory forms [5]. Gingipains hinder the host immune response by degrading complement proteins and cytokines [6,7]. Furthermore, gingipains degrade cell adhesion molecules and cause cell detachment and death due to the loss of intercellular connections [6].

*P. gingivalis* produces LPS and lipoproteins which induce inflammatory responses through activation of Toll-like receptors (TLRs) [8]. LPS is a major component of the outer membrane and is recognized via TLR4, whereas lipoproteins are sensed by TLR2 [9,10]. These TLR ligands activate the downstream signal transduction pathways such as nuclear factor kappa-B (NF-κB) and mitogen-activated protein kinase and induce production of inflammatory cytokines and chemokines [3]. *P. gingivalis* lipoproteins are potent inducers of cytokine production in human gingival fibroblasts compared with the *P. gingivalis* LPS [11]. Thus, not only *P. gingivalis* but also its virulence factors are likely important in the pathogenesis of periodontal disease.

Ozone gas is a powerful oxidant known to inactivate bacteria, viruses, fungi, yeasts, and protozoa [12]. As for bacteria, ozone gas damages bacterial cytoplasmic membrane or cell wall and modifies intracellular contents including protein oxidation and loss of organelle function [13]. On the other hand, ozone gas can have harmful effects on the respiratory tract even at concentrations as low as 0.2 ppm [14]. Thus, instead of ozone gas, ozone water has been used as antimicrobial agent. Ozone water is less harmful than ozone gas due to its low volatilization rate [15]. However, ozone dissolved in water is highly unstable and decomposes rapidly in just 1 h or less [16]. For this reason, we focused on ultrafine bubble technology.

Ozone ultrafine bubbles are ozone gas-filled bubbles less than 1 μm in diameter [17]. Ozone ultrafine bubble water (OUFBW) has antimicrobial activity, can inactivate bacterial toxins as well as ozone gas/water, and is stable for 24 h when stored at 4 °C [18,19]. Furthermore, OUFBW production is inexpensive and easy because it can be made from only water and air using generators. Among periodontal bacteria, OUFBW can sterilize *P. gingivalis, Prevotella intermedia, Fusobacterium nucleatum,* and

*Aggregatibacter actinomycetemcomitans* [18,20]. Per a clinical study, ultrasonic subgingival debridement with OUFBW may be a potential supplement for periodontal treatment [21]. However, the mechanisms underlying the bactericidal effect of OUFBW against *P. gingivalis* and its effect on virulence factors have yet to be fully elucidated.

This study aimed to investigate the effects of OUFBW on *P. gingivalis* and its virulence factors. We also investigated the long-term cytotoxicity of OUFBW on human cells.

## Materials and methods

### 2.1 Production of OUFBW

OUFBW was produced using oxygen and distilled water (DW) as described previously, with modifications [22]. Briefly, ozone gas was generated using a dielectric barrier discharge ozone generator (10 g/h) (Futech-Niigata LLC, Niigata, Japan) with 90% of the oxygen supplied by an oxygen concentrator (flow rate: 3 L/min) (UNICOM, Kanagawa, Japan). OUFBW was generated using a nano blender (Futech-Niigata LLC) and circulated in a stainless steel tank. The ozone concentrations of OUFBW were measured using a Digital Pack Test Ozone kit (Kyoritsu Chemical Check Lab, Tokyo, Japan). Air ultrafine bubble water (AUFBW), which contains room air instead of ozone gas, was produced under the same conditions with OUFBW.

### 2.2 Bacterial culture

*P. gingivalis* strain ATCC 33277 was cultured in modified Gifu anaerobic medium (Nissui, Tokyo, Japan) in an anaerobic jar (Mitsubishi Gas Chemical, Tokyo, Japan) using an AnaeroPack™ anaerobic cultivation system (Mitsubishi Gas Chemical) for 24–48 h at 37 °C. The concentration of the bacterial suspension was measured by optical density at 600 nm ($OD_{600}$) using a Smart Spec™ Plus Spectrophotometer (Bio-Rad, Hercules, CA, USA).

### 2.3 Bactericidal assay using OUFBW

The bactericidal activity of OUFBW against *P. gingivalis* was analyzed using standard plating methods [18]. Briefly, 10 µL of bacterial culture with $OD_{600}$ values of 0.5–0.8 was added to 10 mL of OUFBW containing various concentrations (0.25–3.80 ppm) of ozone and incubated for 0–30 s. OUFBW was prepared by diluting 3.80 ppm OUFBW with DW. AUFBW was used as a control. Thereafter, bacterial cultures exposed to DW or these ultrafine bubble waters (UFBWs; OUFBW and AUFBW) were diluted 1:1 with fetal bovine serum to inactivate ozone immediately. The samples were serially diluted with DW and seeded on trypticase soy agar II with 5% sheep blood plates (Becton Dickinson, Tokyo, Japan). The agar plates were incubated under anaerobic conditions at 37 °C for 7–14 days.

### 2.4 Transmission Electron Microscope (TEM) observation

Two hundred fifty microliters of *P. gingivalis* strain ATCC 33277 bacterial culture ($OD_{600}$ = 1.2) was added to 49.75 mL of AUFBW, OUFBW (2.87 ppm), or DW and incubated for 1 min. The bacterial cells were harvested by centrifugation at 10,000 × *g* for 15 min and fixed with 2% glutaraldehyde in phosphate buffer for 30 min at 4 °C. Thereafter, the samples were centrifuged again and re-fixed with 2% glutaraldehyde. Sample preparation and TEM observation were performed by Filgen, Inc (Nagoya, Japan) [18]. Images of each group at 20,000 × magnification is shown. Additionally, MVs in each of ten TEM images at 10,000 × magnification were counted.

### 2.5 Gingipain activity assay

To determine gingipain activity, an assay was performed using gingipain-specific fluorogenic substrates: Boc-Phe-Ser-Arg-MCA (Peptide Institute Inc, Osaka, Japan), specific for Rgp, and Z-His-Glu-Lys-MCA (Peptide Institute Inc), specific for Kgp. Cultured *P. gingivalis* samples were centrifuged for 10 min at 12,000 × *g*. Thereafter, the culture supernatant was

mixed with AUFBW or 3–4 ppm OUFBW at various dilution ratios and incubated at room temperature for 1 h, followed by sonication for 30 min to deactivate ozone [23]. Ten microliters of each supernatant sample was added to black flat-bottom 96 well plates (Thermo Fisher Scientific, Waltham, USA). Gingipain substrates (0.01 mM, 50 μL) were added to the samples and incubated for 1 h. Plates were read on a NivoS multimode plate reader (PerkinElmer, Waltham, MA, USA) at an excitation wavelength of 380 nm and an emission wavelength of 460 nm.

## 2.6 Silver staining

Three-micrograms of recombinant RgpA (rRgpA; CUSABIO, Houston, TX, USA), rRgpB (CUSABIO), or rKgp (CUSABIO) were added to 500 μL of 3–4 ppm OUFBW or AUFBW, respectively, and incubated for 5 min at 37 °C. The mixture was concentrated using Amicon ultra-0.5 centrifugal filter devices (Merck, Darmstadt, Germany). Thereafter, the mixture was separated by standard sodium dodecyl-sulfate polyacrylamide gel electrophoresis (SDS-PAGE) using a 12% acrylamide gel (Bio-Rad). The resulting gels were silver-stained using the Pierce™ Silver Stain Kit (Thermo Fisher Scientific).

## 2.7 Western blotting

To analyze the effect of OUFBW on gingipain degradation, *P. gingivalis* was cultured for 24–48 h. The samples were centrifuged for 10 min at 12,000 × *g,* and the supernatant was collected. One hundred microliters of the culture supernatant were mixed with 900 μL of 3–4 ppm OUFBW or AUFBW and incubated for 1 h at 37 °C. To analyze the inhibitory effect of OUFBW on gingipain-induced degradation of E-cadherin and IL-6, 100 μL of *P. gingivalis* culture supernatant was mixed with 900 μL of 3–4 ppm OUFBW in the presence or absence of the gingipain inhibitors KYT-1 or KYT-36, and incubated for 1 h at 37 °C. Thereafter, 9 μL of the mixtures was incubated with rE-cadherin (2 ng; Abcam, Cambridge, UK) or rIL-6 (30 ng; Thermo Fisher Scientific) for 2 h at 37 °C. The samples in both experiments were separated by SDS-PAGE using 12% acrylamide gel and transferred to PVDF membranes. The membranes were blocked with 5% skim milk in Tris-buffered saline containing 0.05% Tween-20 (TBST), rinsed with TBST, and incubated with anti-RgpA antibody (CUSABIO, Cat# CSB-PA338957LA01PQP, 1:1000), anti-RgpB antibody (CUSABIO, Cat# CSB-PA310587LA01EYA, 1:1000), anti-Kgp antibody (MyBioSource, Inc., San Diego, CA, USA, Cat# MBS7103929, 1:1000), anti-E-cadherin antibody (Thermo Fisher Scientific, Cat# 20874-1-AP, 1:1000), or anti-IL-6 antibody (Abcam, Cat# ab6672, 1:1000) overnight at 4 °C, followed by incubation with a HRP-conjugated anti-rabbit IgG antibody (Cell Signaling Technology, Beverly, MA, USA, Cat# 7074, 1:3000). ECL Select Western Blotting Detection Reagent (Cytiva, Uppsala, Sweden) and ImageQuant LAS-4000 Mini (Fujifilm, Tokyo, Japan) were used to detect targeted proteins. The signal intensity was quantified using Image Studio software (Ver6.0.0; LI-COR Biosciences, Nebraska, USA).

## 2.8 Secreted embryonic alkaline phosphatase (SEAP) activity assay

*P. gingivalis* LPS (standard grade; InvivoGen, San Diego, CA, USA) was mixed with 3–4 ppm OUFBW or AUFBW to a final concentration of 1 μg/mL and incubated for 1 h at 37 °C, followed by sonication for 30 min to deactivate ozone. HEK-Blue cells expressing human TLR2 (HEK-hTLR2), HEK-hTLR4 and HEK-null2 cells (InvivoGen) were cultured at 37 °C in 5% $CO_2$. The cells were suspended in HEK-Blue Detection medium (InvivoGen), which contains a SEAP substrate and allows for monitoring changes in SEAP levels, and then seeded at a density of $5.0 × 10^4$ cells/180 μL. Then, 20 μL of the LPS-UFBW mixture was added to the wells and incubated for 16 h in 5% $CO_2$. SEAP activity was measured using a Multiskan FC microplate photometer at 620 nm.

## 2.9 Cell viability assay

The human gingival epithelial cell line Ca9-22 (RIKEN Cell Bank, Ibaraki, Japan) was cultured in Minimum Essential Medium α (MEM; Thermo Fisher Scientific), supplemented with 10% fetal bovine serum (Japan Bio Serum, Hiroshima,

 

Japan), 100 U/mL penicillin, and 100 µg/mL streptomycin (FUJIFILM Wako Pure Chemical, Osaka, Japan) at 37 °C in a humidified atmosphere containing 5% $CO_2$. These cells were seeded at a density of $1.0 \times 10^5$ cells/100 µL in 96-well plates and incubated for 24 h. OUFB-RPMI and AUFB-RPMI were prepared by diluting 10 × RPMI (Sigma-Aldrich, Steinheim, Germany) with OUFBW and AUFBW, respectively. Thereafter, cells were exposed to 100 µL of 3.60 ppm OUFB-RPMI, AUFB-RPMI, or 0.1% Triton X-100 for 1–12 h. After treatment, the cells were carefully washed with PBS to remove any residual activity of the reagents. Cell viability was assessed using the MTT [3-(4,5-dimethythiazol-2-yl)-2,5-diphenyltetrazolium bromide] assay. The absorbance of the colored solution was measured spectrophotometrically at 571 nm using a microplate reader (Multiskan FC, Thermo, Waltham, MA, USA).

### 2.10 Statistical analysis

Data were analyzed by one-way or two-way analysis of variance (ANOVA) with Dunnett's or Tukey's multiple-comparison tests using GraphPad Prism version 10.4.2 (GraphPad Software Inc., La Jolla, CA, USA).

## Results

### UFBW exerts bactericidal effects against *P. gingivalis*

We first investigated the bactericidal effects of UFBW against *P. gingivalis*. *P. gingivalis* was exposed to OUFBW (0.25–3.77 ppm) or AUFBW for 30 s followed by colony counting. *P. gingivalis* colonies were not detected after exposure to 1.43 or 3.77 ppm OUFBW (Fig 1A). Additionally, OUFBW (0.25 and 0.81 ppm) and AUFBW significantly decreased colony numbers by approximately 45%, 99%, and 67% respectively, compared to the DW-treated control group. To determine the minimum exposure time required for UFBW to exert a bactericidal effect, *P. gingivalis* was exposed to AUFBW or OUFBW (1.25 ppm or 3.80 ppm) for 0–30 s (Fig 1B). Fig 1C shows that no colonies were detected after a 3-s exposure to OUFBW. Exposure to AUFBW also significantly reduced *P. gingivalis* colonies. Exposure of AUFBW for 30 s reduced the number of colonies to 23.8% compared to the DW group.

### UFBW treatment causes morphologic changes of *P. gingivalis* and decreases the number of membrane vesicles

To investigate the bactericidal mechanism of UFBWs, we evaluated the morphological changes in *P. gingivalis* using TEM. Initially, *P. gingivalis* was exposed to 3.85 ppm OUFBW. However, we could detect only a small amount of cell pellets compared with those in the DW-treated group. Therefore, we reduced the ozone concentration to 2.87 ppm by diluting with DW. We observed that *P. gingivalis* maintained the continuous cell wall structure in the DW-treated group (Fig 2A). In the DW-treated group, numerous MVs were observed surrounding the cells (Fig 2a). Additionally, TEM analysis showed that AUFBW created crevices, most likely between the cell wall and the cytoplasm (Fig 2B). In the AUFBW-treated group, the number of MVs around the cells was reduced compared with that in the DW-treated group (Fig 2b). However, OUFBW damaged the cell wall and inner membrane, leading to cellular content leakage, in addition to creating crevices between the cell wall and the cytoplasm (Fig 2C). In the OUFBW-treated group, the damaged cell wall and inner membrane resulted in morphological changes within the cell (Fig 2c). Additionally, fewer MVs were observed compared with those in the DW-treated group. We counted the number of MVs per *P. gingivalis* and found that the number of MVs in the OUFBW and AUFBW groups was significantly decreased by approximately 80% and 50%, respectively, compared with that in the DW group (Fig 2D).

### UFBW degrades and inactivates gingipains

We investigated the effect of UFBW on the activities of gingipains. *P. gingivalis* culture supernatant was exposed to OUFBW (3.50 ppm) or AUFBW for 60 min, followed by a gingipain activity assay. At a UFBW: supernatant ratio of 500:500, OUFBW and AUFBW significantly decreased Rgp activity by 67% and 55%, respectively, compared with the DW group

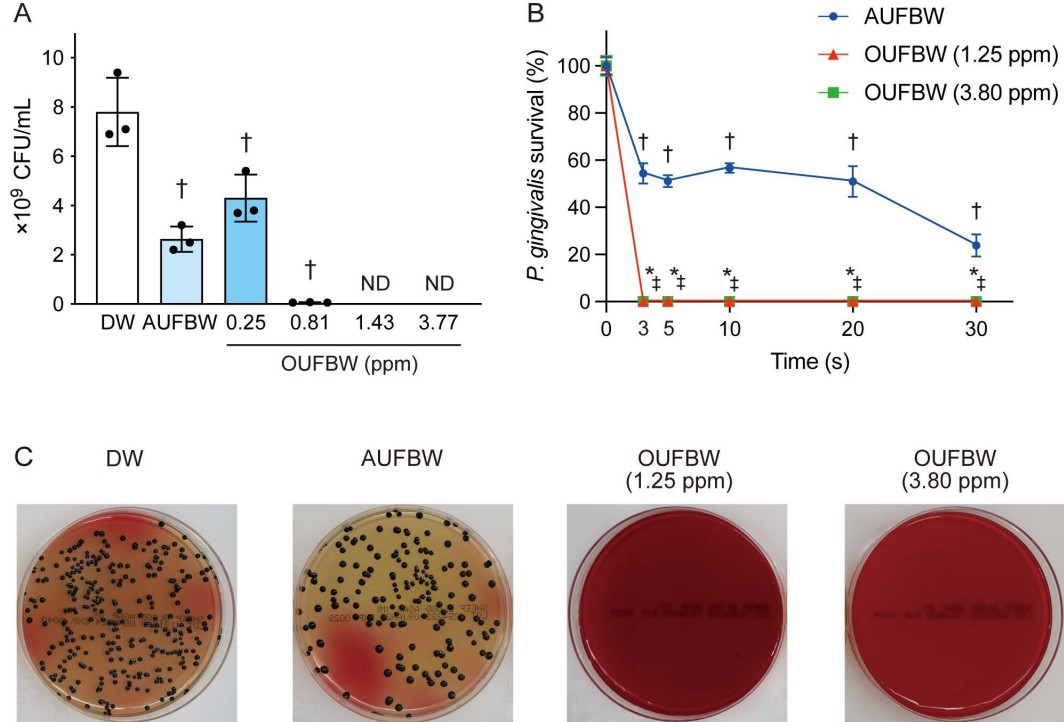

**Fig 1. Ozone ultrafine bubble water exhibits bactericidal effect against *Porphyromonas gingivalis*.** *P. gingivalis* strain ATCC 33277 was exposed to ozone ultrafine bubble water (OUFBW) (0.25–3.8 ppm ozone) or air ultrafine bubble water (AUFBW) for 0–30 s. (A) The bacterial load of *P. gingivalis* was determined by colony count. Data are presented as means ± SD of triplicate experiments and were evaluated by one-way analysis of variance with Dunnett's multiple-comparisons tests. †, significant difference compared to distilled water (DW) group at $P < 0.05$. ND stands for not detected and indicates a result below the detection limit {< $10^2$ colony forming units (CFU) per milliliter}. (B) Time-dependent survival curve of *P. gingivalis* exposed to OUFBW or AUFBW. The data are presented as relative values, with the DW group set to 100. Data are presented as means ± SD of triplicate experiments and were evaluated by two-way analysis of variance with Tukey's multiple-comparisons test. †, significant difference compared to DW group at $P < 0.05$. * and ‡ indicate results below the detection limit within OUFBW (1.25 and 3.80 ppm) group (< $10^2$ CFU per milliliter), respectively. (C) Blood agar plate images of the 3-s exposure groups in Fig 1B were shown.

(Fig 3A, left). Additionally, at UFBW: supernatant ratio of 990:10, OUFBW almost completely inactivated Rgp activity, whereas AUFBW significantly decreased Rgp activity by 79% (Fig 3A, right). Kgp activity was significantly decreased by 89% and 86% following exposure to OUFBW and AUFBW at a UFBW: supernatant ratio of 500:500, respectively (Fig 3B left). OUFBW induced almost complete inactivation of Kgp activity at a UFBW: supernatant ratio of 990:10 (Fig 3B right), whereas AUFBW significantly decreased Kgp activity by 87%. Both Rgp and Kgp activities were decreased even after 30-s exposure of *P. gingivalis* culture supernatant to OUFBW (S1 Fig). These data indicate that OUFBW can inactivate gingipains more effectively than AUFBW at a higher dilution ratio.

Our previous study indicates that OUFBW degrades various bacterial protein toxins [22]. Therefore, we further investigated the ability of UFBW to degrade gingipains. rRgpA, rRgpB, or rKgp were added to OUFBW or AUFBW and subjected to SDS-PAGE followed by silver staining. All these protein bands almost disappeared after exposure to 3.50 ppm OUFBW. Additionally, the band intensity of rRgpA and rRgpB appeared to have decreased after exposure to AUFBW (Fig 4A and B), whereas that of rKgp showed little change (Fig 4C). We next exposed *P. gingivalis* culture supernatants to OUFBW (>3 ppm) or AUFBW, followed by western blotting using anti-RgpA, -RgpB and -Kgp antibodies. After exposure to AUFBW, the band intensities of RgpA and Kgp significantly decreased compared to the DW group (Fig 4D and 4F), while that of RgpB did not (Fig 4E). Additionally, exposure to OUFBW significantly

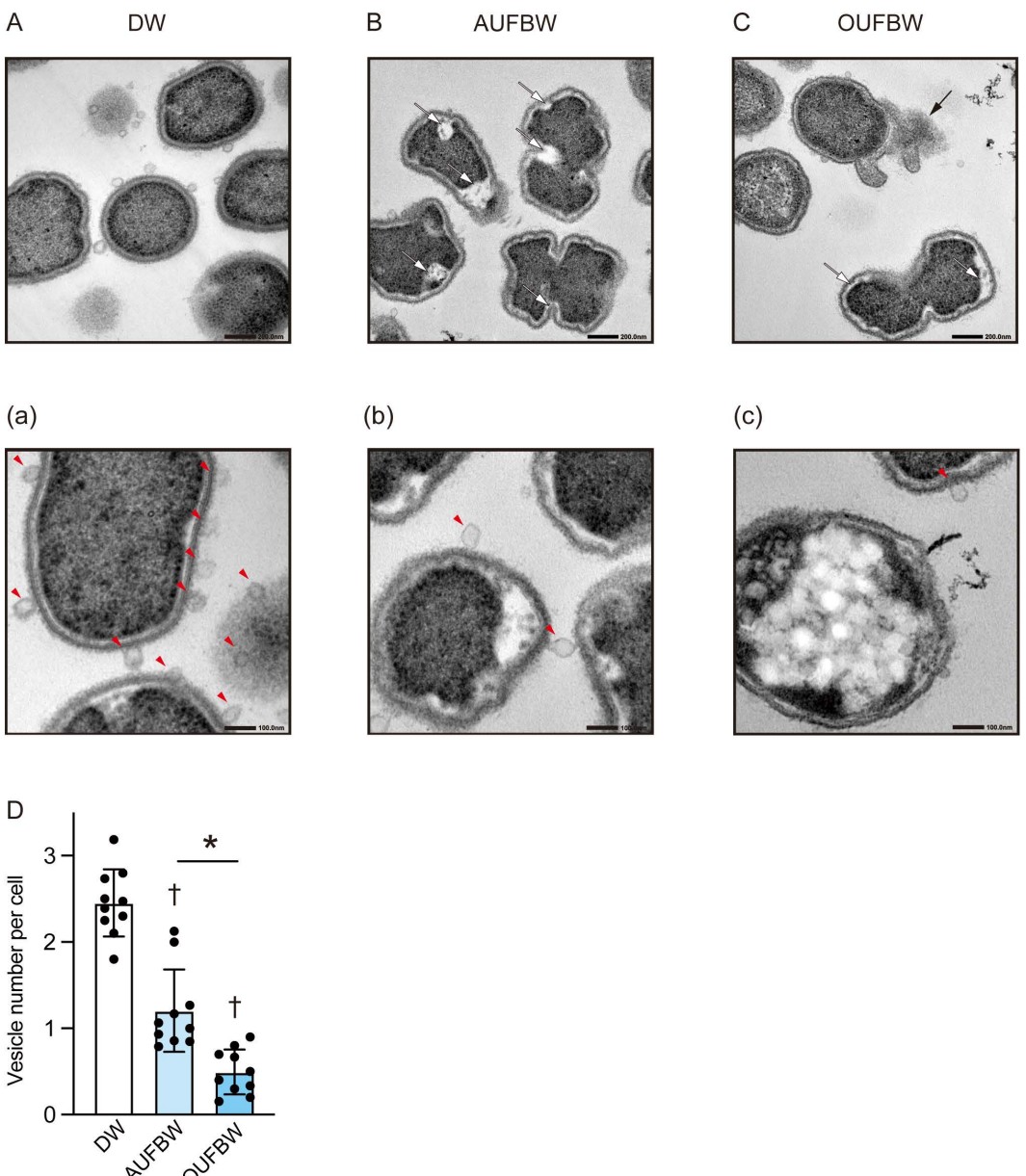

**Fig 2. Transmission electron microscopy of *Porphyromonas gingivalis* exposed to ultrafine bubble water.** Representative transmission electron micrograghs of *P. gingivalis* after 60-s exposure to (A, a) distilled water (DW), (B, b) air ultrafine bubble water (AUFBW), or (C, c) 2.87 ppm ozone ultrafine bubble water (OUFBW). White arrows, black arrows, and red arrowheads indicate crevices between the cell wall and the cytoplasm, cell content leakage, and membrane vesicles, respectively. Scale bar represents (A, B, C) 200 nm and (a, b, c) 100 nm. (D) The number of membrane vesicles per bacterial cell is shown. The data represented the means ± SD of 10 microscopic fields and were evaluated using one-way analysis of variance with Tukey's multiple comparisons tests. †, significant difference compared with the DW group at $P < 0.05$. *, significant difference between the indicated groups at $P < 0.05$.

decreased the band intensities of RgpA, RgpB, and Kgp compared to both the DW and AUFBW groups (Fig 4D–4F). These findings suggest that both OUFBW and AUFBW partially degrade gingipains in the *P. gingivalis* supernatant, and OUFBW is more efficient.

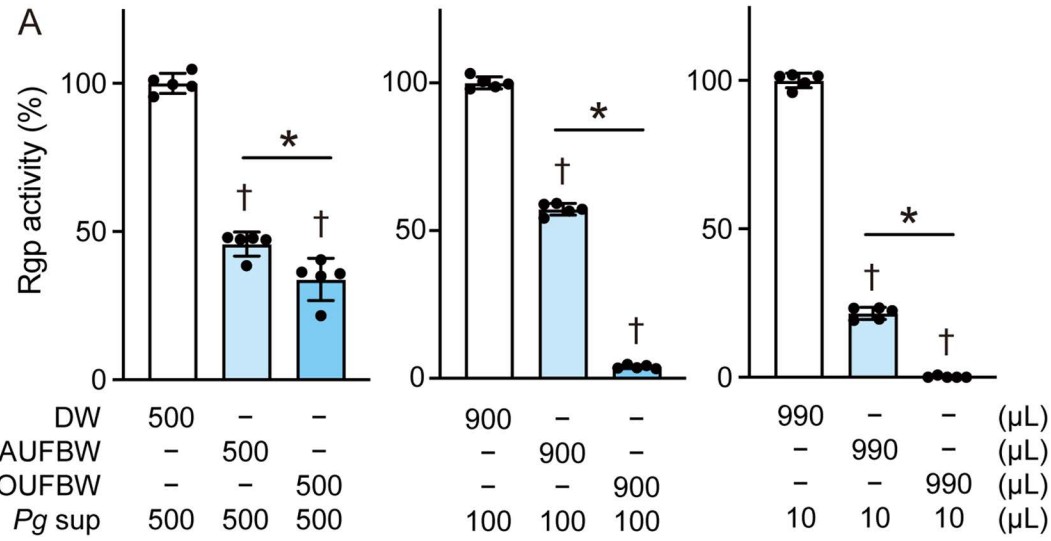

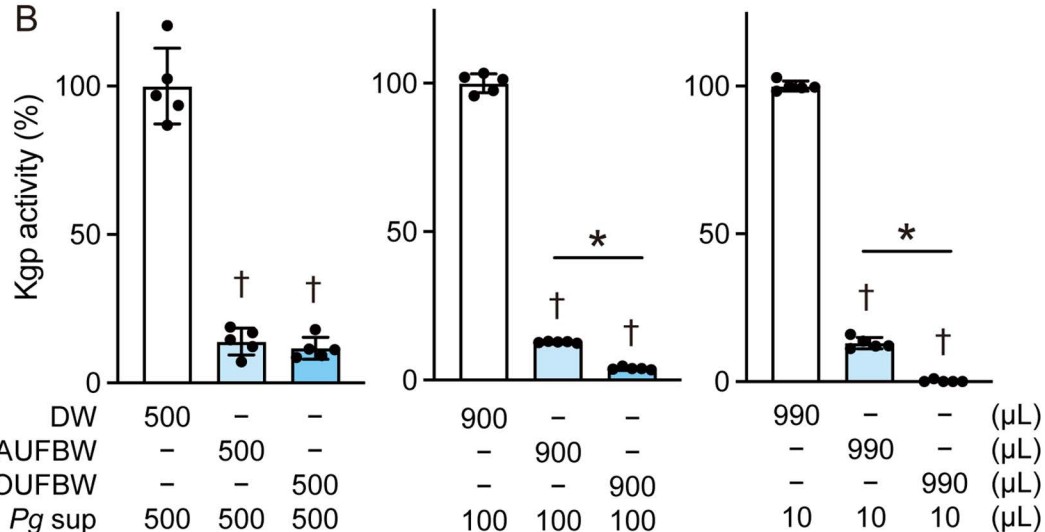

**Fig 3. Ozone ultrafine bubble water decreases gingipain activities in _Porphyromonas gingivalis_ culture supernatant.** _P. gingivalis_ culture supernatant (_Pg_ sup) was exposed to distilled water (DW), air ultrafine bubble water (AUFBW), or 3.52 ppm ozone ultrafine bubble water (OUFBW) for 60 min. (A) Rgp and (B) Kgp activities were determined using the Rgp- and Kgp-specific substrates, respectively. The values of OUFBW and AUFBW group are normalized against that of DW group. The data represented means ± SD of quintuplicate experiments and were evaluated by one-way analysis of variance with Tukey's multiple comparisons tests. †, significant difference compared to DW group at $P < 0.05$. *, significant difference between the indicated groups at $P < 0.05$.

### Exposure of _P. gingivalis_ culture supernatant to OUFBW inhibits degradation of E-cadherin and IL-6 by gingipains

Gingipains are reportedly involved in the degradation of adherents junction protein E-cadherin, leading to the invasion of _P. gingivalis_ into host tissues [24]. Moreover, gingipains cleave and inactivate IL-6 [25]. We investigated whether UFBW-induced inactivation of gingipains results in decreased degradation of E-cadherin and IL-6. We added OUFBW, AUFBW, or DW to _P. gingivalis_ culture supernatant and incubated it for 60 min followed by incubation with rE-cadherin

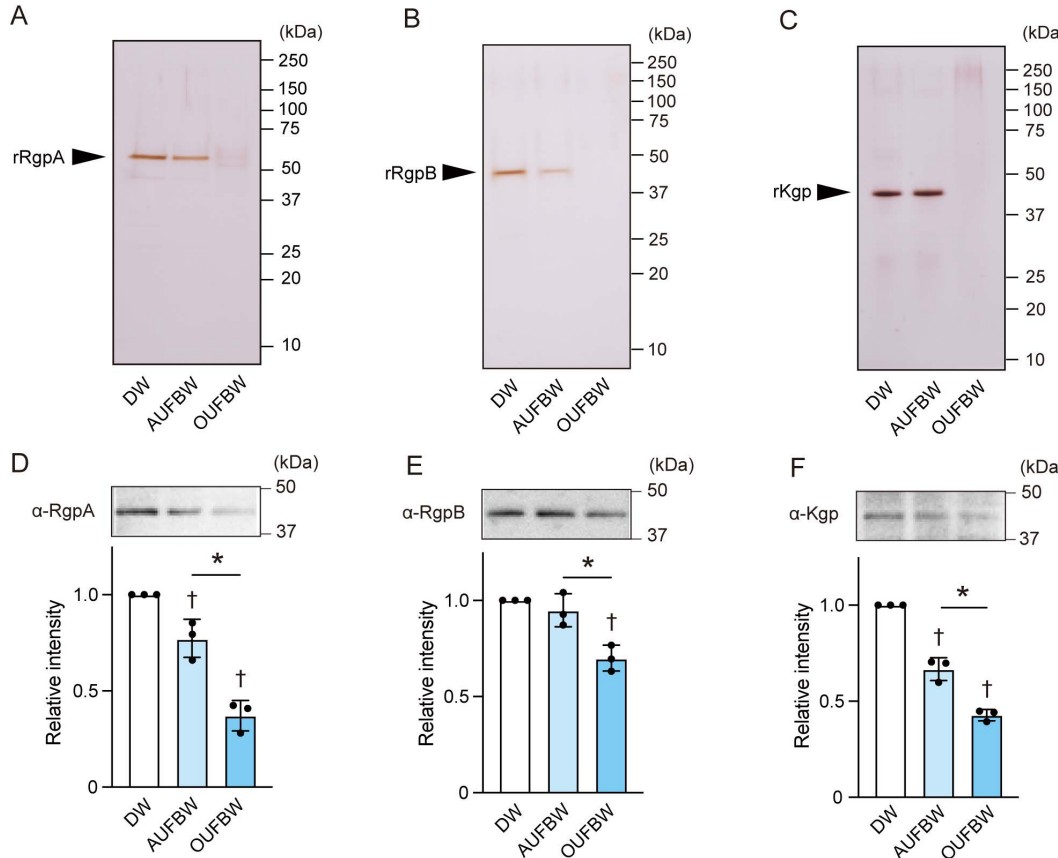

**Fig 4. Ozone ultrafine bubble water degrades gingipains.** (A) Recombinant RgpA (rRgpA), (B) rRgpB, and (C) rKgp were added to distilled water (DW), air ultrafine bubble water (AUFBW), or 3.50 ppm ozone ultrafine bubble water (OUFBW) followed by SDS-PAGE and silver staining. (D–F) *P. gingivalis* culture supernatant was exposed to DW, AUFBW, or OUFBW and incubated for 60 min. (D) RgpA, (E) RgpB, and (F) Kgp were detected by western blotting. Representative images are shown. Intensities of western blotting signals for RgpA, RgpB, and Kgp were quantified by densitometry. The data represented means ± SD of triplicate experiments and were evaluated by one-way analysis of variance with Tukey's multiple comparisons test. †, significant difference compared to DW group at $P < 0.05$. *, significant difference between the indicated groups at $P < 0.05$.

or rIL-6 for 120 min in the presence or absence of gingipain inhibitors KYT-1 or KYT-36. Western blot analysis revealed that *P. gingivalis* culture supernatant + DW group almost completely degraded both rE-cadherin and rIL-6 (Fig 5A and 5B; lane 2). There was no significant difference in band intensity between the DW and AUFBW groups (lane 3). However, exposure of *P. gingivalis* culture supernatant to OUFBW significantly inhibited the degradation of rE-cadherin and rIL-6 by gingipains (lane 4). Additionally, degradation of rE-cadherin and rIL-6 was partially inhibited in the presence of KYT-1 or KYT-36 and more strongly inhibited in the presence of both KYT-1 and KYT-36. These results indicate that exposure of *P. gingivalis* culture supernatants to OUFBW inhibits the degradation of E-cadherin and IL-6 by gingipains.

## Pretreatment of *P. gingivalis* LPS with UFBW inhibits TLR2 activation without affecting TLR4 signaling

A previous study showed that OUFBW suppressed TLR2 activation by Pam3CSK4, a synthetic lipopeptide. While LPS, the main component of the outer membrane, is predominantly recognized by TLR4, *P. gingivalis* LPS induces the production of inflammatory cytokines via both TLR2 and TLR4 because of contaminating lipoproteins [11,26–28]. To investigate whether UFBW inhibits the proinflammatory activities of *P. gingivalis* LPS, we performed SEAP assay using HEK-hTLR2

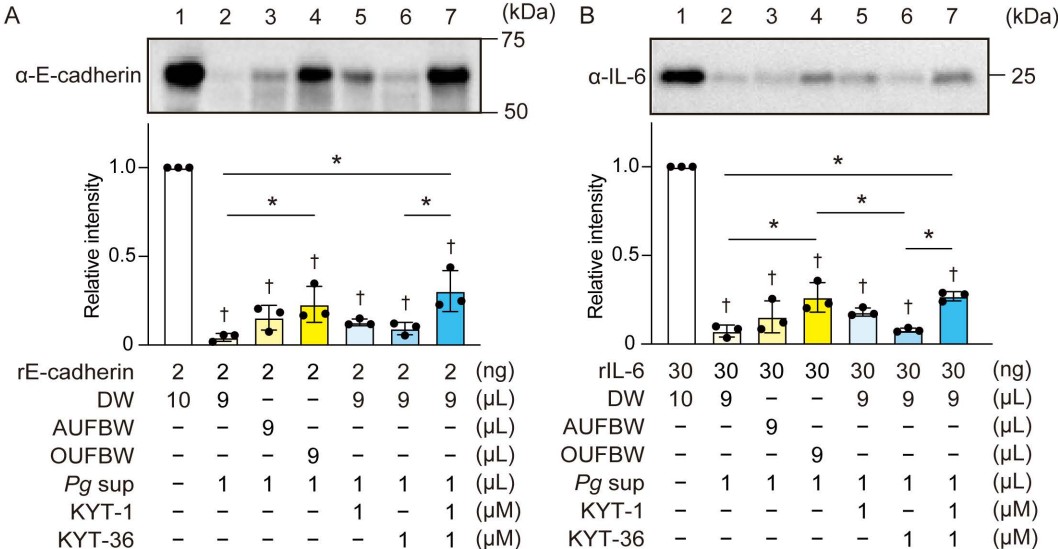

**Fig 5. Pretreatment of *Porphyromonas gingivalis* supernatant with ozone ultrafine bubble water inhibits degradation of E-cadherin and IL-6.** (A, B) *P. gingivalis* culture supernatant was exposed to distilled water (DW), air ultrafine bubble water (AUFBW), or ozone ultrafine bubble water (OUFBW) for 60 min followed by incubation with recombinant E-cadherin (rE-cadherin) or rIL-6 for 120 min in the presence or absence of gingipain inhibitors, KYT-1 or KYT-36. (A) rE-cadherin and (B) rIL-6 were detected by western blotting. Representative images are shown. Intensities of western blotting signals of α-E-cadherin and α-IL-6 were quantified by densitometry. The data represented means ± SD of triplicate experiments and were evaluated by one-way analysis of variance with Tukey's multiple comparisons tests. †, significant difference compared to DW group at $P < 0.05$. *, significant difference between the indicated groups at $P < 0.05$.

or HEK-hTLR4 cells. HEK-Blue cells release SEAP into the cell culture medium in a NF-κB-dependent manner. We stimulated these cells with UFBW-pretreated *P. gingivalis* LPS. Pretreatment of *P. gingivalis* LPS with OUFBW significantly reduced the induction of SEAP activity in HEK-hTLR2 cells compared to that of the DW-pretreated group (Fig 6A). Neither OUFBW nor AUFBW affected the TLR4-stimulatory activity of *P. gingivalis* LPS (Fig 6B). *P. gingivalis* LPS did not induce SEAP release from HEK-null2 cells (which possess the SEAP reporter gene but lack transfected TLRs) (Fig 6C). These findings suggest that OUFBW can degrade lipoproteins and cannot degrade LPS.

### UFBW have low cytotoxicity toward human cell line Ca9-22

The cytotoxicity of UFBW to the human gingival epithelial cell line Ca9-22 was analyzed to assess their potential use as disinfectants for dental instruments and in the oral cavity. Exposure of the cells to OUFB-RPMI (ozone concentration 3.60 ppm) and AUFB-RPMI for 1 h did not significantly reduce cell viability. After 12 h of exposure, the viability of OUFB-RPMI- and AUFB-RPMI-treated cells was 85% and 82%, respectively (Fig 7). These results indicate that UFBW exhibits minimum cytotoxicity at 1-h exposure.

### Discussion

In this present study, OUFBW induced cytoplasmic leakage in *P. gingivalis*, suggesting that both the cell wall and inner membrane were disrupted. The cell wall of Gram-negative bacteria comprises an outer membrane situated above a thin peptidoglycan layer [29]. The outer membrane is a lipid-protein bilayer, mainly composed of phospholipids, LPS, and membrane proteins [29]. Peptidoglycan is a heteropolymer of glycan strands crosslinked by peptides [30]. In the present study, LPS was not inactivated by OUFBW, suggesting that OUFBW does not target its active sites. This finding suggested that the outer membrane can still be disrupted by OUFBW even without LPS destruction. The inner membrane is

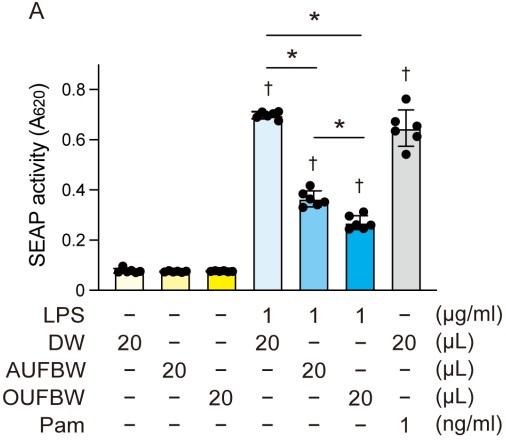

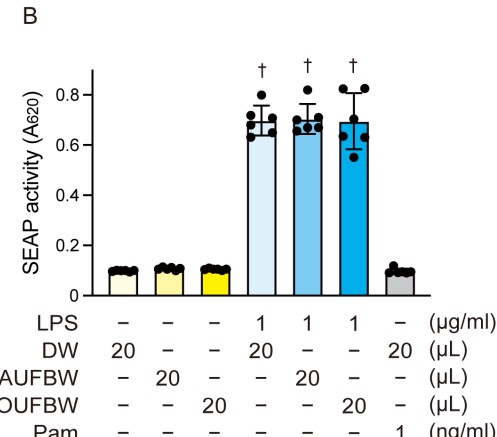

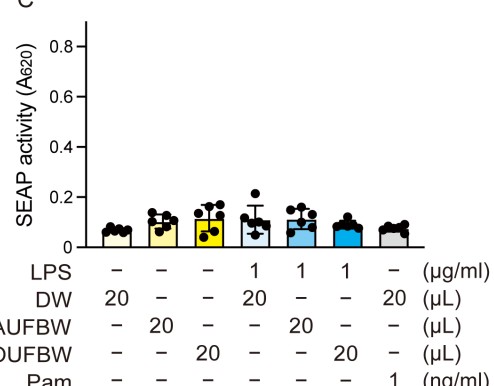

**Fig 6. *Porphyromonas gingivalis* LPS pretreated by ozone ultrafine bubble water induces decreased TLR2 activation.** *P. gingivalis* LPS was added to distilled water (DW), air ultrafine bubble water (AUFBW), or 3.03 ppm ozone ultrafine bubble water (OUFBW). (A) HEK-Blue human Toll-like receptor 2 (hTLR2), (B) HEK-Blue hTLR4, and (C) HEK-Blue null2 cells were stimulated with the LPS. Pam3CSK4 (Pam), a TLR2 agonist, was used as the control. Secreted alkaline phosphatase (SEAP) levels were quantified using spectrophotometry at 620 nm. The data represent means ± SD of sextuplicate experiments and were evaluated by one-way analysis of variance with Tukey's multiple comparisons test. †, significant difference compared to DW group at $P < 0.05$. *, significant difference between the indicated groups at $P < 0.05$.

composed of phospholipids and membrane proteins. The sulfhydryl and amino acid groups of proteins have been reported to be oxidized by ozone gas [31]. Accordingly, we considered that OUFBW damaged these specific amino acid groups of proteins in the cell wall and inner membrane. The bactericidal mechanism of OUFBW will be more clearly understood by investigating its effects on peptidoglycan and bacterial lipid components.

OUFBW almost completely degraded recombinant gingipains, which represent 21–33% of the total amino acid sequence of the original gingipains. In contrast, OUFBW only partially degraded gingipains in *P. gingivalis* culture supernatant. Recombinant gingipains may be more susceptible to degradation by OUFBW. One possible explanation is that the tertiary structure of recombinant gingipains differs from that of the original gingipains. RgpB in culture supernatant showed less degradation than of RgpA or Kgp. RgpA and Kgp are multi-domain proteins composed of catalytic domains and haemagglutinin/adhesin (HA) domains. RgpB lacks the HA domains and has a protein structure described as a "crooked one-root tooth" [32]. These structural differences among the gingipains may lead to differential susceptibility to OUFBW-mediated protein degradation.

Activation of TLRs causes nuclear translocation of NF-κB and transcription of inflammation-related genes [33]. TLR4 is one of the pattern recognition receptors, which recognizes LPS [28]. However, previous research reported

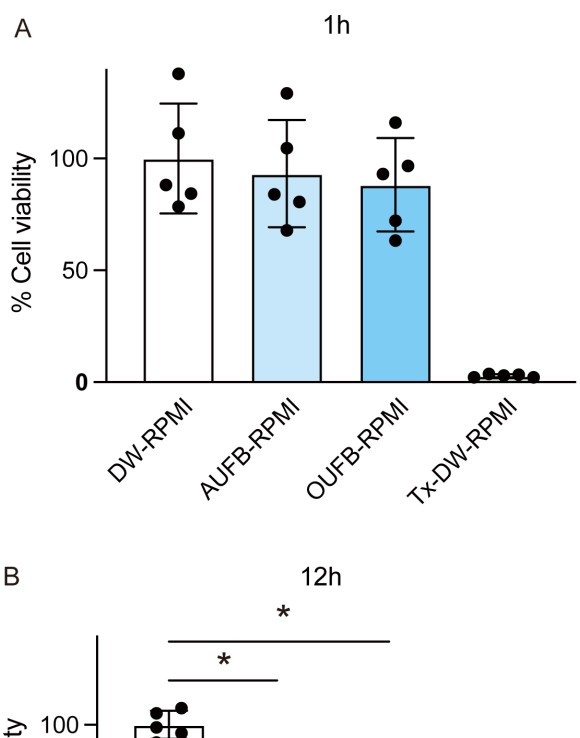

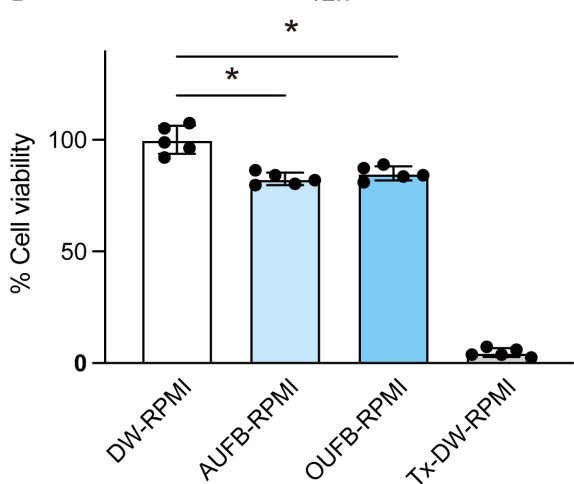

**Fig 7. Ultrafine bubble water exhibits minimal cytotoxic effects on human gingival epithelial cell line after 1-h exposure.** Human gingival cell line Ca9-22 was exposed to RPMI containing distilled water (DW), air ultrafine bubble water (AUFB), 3.60 ppm ozone ultrafine bubble water (OUFB), or 0.1% Triton X-100 (Tx) for (A) 1 h or (B) 12 h. MTT (3-(4,5-Dimethyl-2-thiazolyl)-2,5-diphenyl-2H-tetrazolium bromide) assays were performed to determine cell viability. The data represented the means ± SD of quintuplicate experiments and were evaluated by one-way analysis of variance with Dunnett's multiple comparisons test. *, significant difference compared to DW-RPMI group at $P < 0.05$.

that both TLR2 and TLR4 are involved in the recognition of *P. gingivalis* LPS [28]. Whether *P. gingivalis* LPS activates TLR2 has been discussed and it was eventually revealed that contaminant lipoproteins in the LPS activate TLR2 [11]. Standard *P. gingivalis* LPS is commercially available as well as ultrapure LPS which undergoes enzymatic treatment to remove lipoproteins [28]. We used standard *P. gingivalis* LPS, which activates both TLR2 and TLR4, to analyze the effects of OUFBW on LPS and lipoproteins simultaneously. Pretreating standard *P. gingivalis* LPS with OUFBW led to reduced TLR2 activation in HEK-TLR2 cells, but it did not affect TLR4 activation in HEK-TLR4 cells compared with DW-pretreated group. These data indicate that OUFBW inactivates lipoproteins but does not inactivate LPS. Additionally, previous studies reported that OUFBW degrades bacterial protein toxins, such as leukotoxin, pneumolysin, and staphylococcal enterotoxin A, whereas it does not degrade LPS from both *A.*

*actinomycetemcomitans* and *Escherichia coli* [22,34]. These studies emphasize that OUFBW degrades proteins such as gingipains and lipoproteins.

Our study showed that OUFBW not only sterilizes *P. gingivalis* but also reduces its MVs. MVs are about 50–250 nanometers in diameter [29], and are mainly composed of bacterial outer membrane proteins. MVs are produced by blebbing during growth or explosive cell lysis and stimulate the production of proinflammatory cytokines in macrophages more effectively than bacterial cells [35–37]. To our knowledge, this study is the first to show that OUFBW reduces MVs. We propose two mechanisms for OUFBW-induced MVs reduction. First, sterilization of *P. gingivalis* by OUFBW leads to reduced MVs production. Second, OUFBW may damage both MVs and bacterial outer membrane because they are composed of the same constituents. It was reported that *P. gingivalis* MVs contain major virulence factors such as gingipains, lipoproteins, and LPS [37]. Among these, OUFBW inactivated gingipains and lipoproteins. Taken together, OUFBW-induced reduction of MVs may reduce bacterial virulence.

Unexpectedly, AUFBW also damaged the membranes of *P. gingivalis,* reduced the number of MVs, and partially inactivated gingipains and lipoproteins, although to a lesser extent than OUFBW. We suspect two causes of bacterial cell damage induced by AUFBW: bubble collapse and oxygen. Per previous reports, the destruction of bubbles may be the cause of bacterial cell damage [38,39]. The underlying mechanisms may be bubble collapse-induced pressure waves and hydroxyl radicals which can oxidize substances [40]. As for oxygen, hydroxyl radical are formed when the oxygen concentrations in bacterial cells increase [41,42]. As an obligate anaerobic microorganism, *P. gingivalis* does not possess the antioxidant activity necessary to detoxify hydroxyl radicals [41]. As a result, hydroxyl radicals exhibit toxicity to *P. gingivalis* cells [43]. Indeed, a decrease in *P. gingivalis* cell viability was observed as the percentage of oxygen increased in gas phase [44]. These results suggest that ultrafine bubble collapse or oxygen may be the cause of bacterial cell damage and MVs reduction. Furthermore, hydroxyl radicals may also be involved in the action of AUFBW on proteins such as gingipains and lipoproteins because it is known that hydroxyl radicals can oxidize proteins [41]. For example, hydroxyl radicals generating through metal-catalyzed oxidation systems can degrade albumin [45]. These results suggest that hydroxyl radicals induced by AUFBW may also inactivate proteins including gingipains and lipoproteins.

We showed that 3.60 ppm OUFB-RPMI and AUFB-RPMI exhibited low cytotoxicity toward Ca9-22 cells at 1 h, whereas cell viability was reduced after 12-h exposure. Clinical use of UFBW as disinfectants in the oral cavity is expected to be shorter than 1 h. Disinfectants used in the oral cavity are applied for 30–60 s, and 0.2% chlorhexidine (CHX) is widely used as a mouthrinse [46]. According to a previous study, OUFBW is less harmful to buccal and gingival epithelial cells compared with 0.2% CHX [20]. These findings indicate that OUFBW and AUFBW are safer agents than CHX *in vitro*. The bactericidal effects and safety of OUFBW and AUFBW should continue to be investigated *in vivo*.

## Conclusions

In summary, our study demonstrated that OUFBW sterilizes *P. gingivalis* by damaging the cell walls and inactivating gingipains and lipoproteins. Moreover, OUFBW showed low cytotoxicity against periodontal epithelial cells. These results suggest that OUFBW would be used to disinfect periodontal treatment instruments contaminated with *P. gingivalis* and its virulence factors. Furthermore, we hypothesized that even if OUFBW remained on the dental instruments, it would be minimally invasive to gingival cells, as it was minimally cytotoxic against the human gingival epithelial cell line Ca9-22. Currently, we do not condone using UFBW directly on periodontal tissues; however, we would like to investigate its effect on cleaning the gingival sulcus in the future. At that time, we also plan to perform a cytotoxicity assay using primary gingival cells.

## Supporting information

**S1 Fig. Ozone ultrafine bubble water decreases gingipain activities in *Porphyromonas gingivalis* culture supernatant after 30-s exposure.** *P. gingivalis* culture supernatant (*Pg* sup) was exposed to distilled water (DW), air ultrafine bubble water (AUFBW), or 4.14 ppm ozone ultrafine bubble water (OUFBW) for 30 s. (A) Rgp and (B) Kgp activities were

determined using Rgp- and Kgp-specific substrates, respectively. The values of the OUFBW and AUFBW groups were normalized against that of the DW group. The data represented the means ± SD of quintuplicate experiments and were evaluated using one-way analysis of variance with Tukey's multiple comparisons tests. †, significant difference compared with the DW group at $P < 0.05$. *, significant difference between the indicated groups at $P < 0.05$.
(PDF)

**S2 Fig. Original images of sliver stain gels.** Untreated silver-stained images of Figure. 4A–C are shown. Images were obtained by scanning the gel using an image scanner.
(PDF)

**S3 Fig. Original images of western blotting.** Untreated western blotting membranes are shown. The area framed in red are shown in each figure.
(PDF)

## Acknowledgments

We thank Dr. Yoshihito Yasui, Dr. Rui Saito (Niigata University) and Mr. Tadashi Hiwatashi (Futech-Niigata LLC) for their technical support. We also acknowledge Filgen, Inc for observation of TEM.

## Author contributions

**Conceptualization:** Yutaka Terao.

**Data curation:** Hisanori Domon.

**Formal analysis:** Mana Endo.

**Funding acquisition:** Mana Endo, Hisanori Domon, Satoru Hirayama, Yutaka Terao.

**Investigation:** Mana Endo.

**Methodology:** Mana Endo, Hisanori Domon, Satoru Hirayama, Fumio Takizawa.

**Project administration:** Hisanori Domon.

**Resources:** Mana Endo, Hisanori Domon, Akiomi Ushida.

**Supervision:** Hisanori Domon, Koichi Tabeta, Yutaka Terao.

**Visualization:** Mana Endo.

**Writing – original draft:** Mana Endo.

**Writing – review & editing:** Hisanori Domon, Yutaka Terao.

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
