## [Decision Letter · Decision Letter 0]

7 Aug 2025

Dear Dr. Terao,

Thank you for submitting your manuscript to PLOS ONE. After careful consideration, we feel that it has merit but does not fully meet PLOS ONE’s publication criteria as it currently stands. Therefore, we invite you to submit a revised version of the manuscript that addresses the points raised during the review process.

We look forward to receiving your revised manuscript.

Kind regards,

Geelsu Hwang, Ph.D.

Academic Editor

PLOS ONE

Journal Requirements: 

Reviewers' comments:

Reviewer's Responses to Questions

**Comments to the Author**

1. Is the manuscript technically sound, and do the data support the conclusions?

Reviewer #1: Partly

Reviewer #2: Yes

2. Has the statistical analysis been performed appropriately and rigorously?

Reviewer #1: Yes

Reviewer #2: Yes

3. Have the authors made all data underlying the findings in their manuscript fully available?

Reviewer #1: Yes

Reviewer #2: Yes

4. Is the manuscript presented in an intelligible fashion and written in standard English?

Reviewer #1: No

Reviewer #2: Yes

Reviewer #1: Endo et al. presented the bactericidal effect of the ozone encapsulated in ultra-fine bubble water against periodontal pathogen with their pathogenic factors. Their previous study showed that OUFBW showed efficacy in killing gram-positive bacteria, whereas present study showed the efficacy against gram-negative bacteria along with elucidating the bactericidal mechanism.

I have found a few issues that, once addressed, will improve the manuscript.

Comments,

1. In the 2.3 of the material & methods section, please add the explanation of D.W.-treated group. Why did you use two methods for ozone inactivation, such as FBS and sonication.

2. In the 2.8 of the material & methods section, please spell “SEAP” out.

3. In the 2.10 of the material & methods section, please add “one-way or two-way” before “analysis of variance.”

4. In the “UFBW exerts …” section of results section, please describe “Fig. 1C.”

5. In the Abstract, line 12, and in the conclusion section, you mentioned OUFBW disrupted the P. gingivalis cell membrane or cell wall, although the OUFBW did not damage LPS, component of outer membrane. Please explain this point in the discussion section. Further, in the “UFBW treatment…” section of the results section, it wasn’t clear to me where crevices occurred. Did the crack form between cytoplasm and cell membrane, cell wall, or outer membrane? Furthermore, in the discussion section (lines 4 and 5), you mentioned that OUFBW disrupts the bacterial cell wall of gram-positive bacteria and the cell membrane of gram-negative bacteria. From these descriptions, it seems that OUFBW damages gram-positive and negative bacteria by different ways. Previous your study showed that the detachment of the gram-positive cell wall from the cytoplasm. Please explain these points clearly.

6. The authors showed that the exposure condition of OUFBW for inactivation of gingipains from culture supernatant was 3.5 ppm for 1 hour. This condition seems to be harmful to human use. Were the gingipains inactivated and degraded after 30 sec exposure of OUFBW?　In this regard, you should use human primary gingival epithelial cells for cytotoxicity tests because Ca922 is carcinoma cell line.

7. I think the use of AUFBW has potential as a disinfectant against P. gingivalis.

8. In the discussion section (page 24, line 8 from the bottom), please modify “gingivalis” to “P. gingivalis”.

Reviewer #2: The manuscript entitled “Ozone ultrafine bubble water sterilizes Porphyromonas gingivalis and neutralizes its virulence factors” by Mana Endo showed that Ozone ultrafine bubble water (OUFBW) sterilize P. gingivalis (Pg) by destroying bacterial cells and decreasing the number of Pg OMVs. They also demonstrated that USBW degrades gingipains and inhibits TLR2 signaling, and propose that USBW is an effective treatment for periodontal disease.

The data is clear, and there are no major contradictions with the conclusions. However, the following points could be improved:

Major comments:

1. Figure 1 and 2; In addition to DW, it is necessary to compare with a group that is not exposed to any effects (no-treated Pg).

3. Figure 1C; There is no explanation for Figure 1c. Does this result refer to the effect of UFBW after a certain amount of time?

4. The authors should discuss the mechanism of the bactericidal action of UFBW shown in Fig. 1AB. From Fig. 1D, can we see whether Pg bacteria burst (or do not burst) due to osmotic pressure caused by DW or UFBW?

5. In figure 1D, it is unclear whether the structure indicated by the arrowhead is an OMV. It is necessary to enlarge the image to show the structure that is continuous with the cell membrane of the Pg bacteria. If UFBW degrades OMVs and reduces their number, it may be possible to apply UFBW to OMVs isolated from the culture supernatant.

6. In figure 3, Does OUFBW specifically degrade gingipain? If so, what mechanisms could be involved?

7. Fig. 6 clearly shows that OUFB is not harmful to Ca9-22, so it is reasonable for the authors to propose that OUFBW is clinically effective for periodontal disease treatment. However, figure 1D shows that OUFBW lyses Pg bacteria and releases various physiologically active bacterial contents into the surrounding area. Are these bacterial contents harmful to cells? For example, when OUFBW-treated Pg is applied to Ca9-22, does it cause cellular damage?

Minor comments:

1. The source of the recombinant proteins (rRgpA, rRgpB, rKgp, rE-cadherin, rIL-6) should be indicated in material & methods section.

**Do you want your identity to be public for this peer review?** For information about this choice, including consent withdrawal, please see our Privacy Policy

Reviewer #1: No

Reviewer #2: No

---

## [Author Response · Author response to Decision Letter 1]

16 Sep 2025

Please refer to MS word file titled “Response to Reviewers” for Figures of reviewer2’s comments 1, 4, and 7.

Response to Editor

We thank the Editor and the Reviewers for their critical comments that have helped us to improve our manuscript. As indicated in the responses below, we have considered all these comments and addressed each of them during our revision of the manuscript.

<Comment #1> Please ensure that your manuscript meets PLOS ONE's style requirements, including those for file naming.

<Response> According to the Editor’s comment, we have renamed the file accordingly and ensured that our manuscript and figures adhere to PLOS ONE's style requirements.

<Comment #2> PLOS ONE now requires that authors provide the original uncropped and unadjusted images underlying all blot or gel results reported in a submission’s figures or Supporting Information files. This policy and the journal’s other requirements for blot/gel reporting and figure preparation are described in detail at https://journals.plos.org/plosone/s/figures#loc-blot-and-gel-reporting-requirements and https://journals.plos.org/plosone/s/figures#loc-preparing-figures-from-image-files. When you submit your revised manuscript, please ensure that your figures adhere fully to these guidelines and provide the original underlying images for all blot or gel data reported in your submission. See the following link for instructions on providing the original image data: https://journals.plos.org/plosone/s/figures#loc-original-images-for-blots-and-gels. In your cover letter, please note whether your blot/gel image data are in Supporting Information or posted at a public data repository, provide the repository URL if relevant, and provide specific details as to which raw blot/gel images, if any, are not available. Email us at plosone@plos.org if you have any questions.

<Response> Figure S2 and S3 (in the revised version): According to the editor’s comment, we have submitted the original blot/gel image data as supporting information (Figure S2 and S3).

<Comment #3> If the reviewer comments include a recommendation to cite specific previously published works, please review and evaluate these publications to determine whether they are relevant and should be cited. There is no requirement to cite these works unless the editor has indicated otherwise.

<Response> There were no such comments.

<Comment #4> Please review your reference list to ensure that it is complete and correct. If you have cited papers that have been retracted, please include the rationale for doing so in the manuscript text, or remove these references and replace them with relevant current references. Any changes to the reference list should be mentioned in the rebuttal letter that accompanies your revised manuscript. If you need to cite a retracted article, indicate the article’s retracted status in the References list and also include a citation and full reference for the retraction notice.

<Response> According to the Editor’s comment, we have ensured that our references adhere to PLOS ONE’s style requirements. We have added the following reference to the reference list: 30. Schumann P. Peptidoglycan structure. Methods Microbiol. 38: Elsevier; 2011. p. 101-29.

Response to Reviewer 1

We are grateful to Reviewer 1 for the critical comments and suggestions, which have helped us improve our paper considerably. According to your comments, we performed some additional experiments and added several Figures, while also addressing all these comments and suggestions in the revised version of our paper.

<Comment #1> In the 2.3 of the material & methods section, please add the explanation of D.W.-treated group. Why did you use two methods for ozone inactivation, such as FBS and sonication.

<Response> Lines 110–112 (in the revised version): According to the Reviewer’s comment, we have revised the manuscript as follows: “Thereafter, bacterial cultures exposed to DW or these ultrafine bubble waters (UFBWs; OUFBW, and AUFBW) were diluted 1:1 with fetal bovine serum to inactivate ozone immediately”. Different times are required for the deactivation of ozone using FBS and sonication. In Figure 1 (in the revised version), FBS was used in the experiment because it can deactivate ozone immediately. However, in Figures 3 and 6 (in the revised version), sonication was used because FBS may affect the activities of gingipains. Approximately 30 min are required for the deactivation of ozone by sonication (lines 126–128 in the revised version: “Thereafter, the culture supernatant was mixed with AUFBW or 3–4 ppm OUFBW at various dilution ratios and incubated at room temperature for 1 h, followed by sonication for 30 min to deactivate ozone [23]”).

<Comment #2> In the 2.8 of the material & methods section, please spell “SEAP” out.

<Response> Line 160 (in the revised version): According to the Reviewer’s comment, we have revised the manuscript as follows: “2.8 Secreted embryonic alkaline phosphatase (SEAP) activity assay”.

<Comment #3> In the 2.10 of the material & methods section, please add “one-way or two-way” before “analysis of variance.”

<Response> Line 182 (in the revised version): Based on the reviewer’s comment, we have added a description to the revised manuscript as follows: “Data were analyzed by one-way or two-way analysis of variance (ANOVA) with Dunnett’s or Tukey’s multiple-comparison tests using GraphPad Prism version 10.4.2 (GraphPad Software Inc., La Jolla, CA, USA)”.

<Comment #4> In the “UFBW exerts …” section of results section, please describe “Fig. 1C.”

<Response> Lines 192–193 (in the revised version): According to the Reviewer’s comment, we have revised the manuscript as follows: “Figure 1C shows that no colonies were detected after a 3-s exposure to OUFBW”.

<Comment #5> In the Abstract, line 12, and in the conclusion section, you mentioned OUFBW disrupted the P. gingivalis cell membrane or cell wall, although the OUFBW did not damage LPS, component of outer membrane. Please explain this point in the discussion section. Further, in the “UFBW treatment…” section of the results section, it wasn’t clear to me where crevices occurred. Did the crack form between cytoplasm and cell membrane, cell wall, or outer membrane? Furthermore, in the discussion section (lines 4 and 5), you mentioned that OUFBW disrupts the bacterial cell wall of gram-positive bacteria and the cell membrane of gram-negative bacteria. From these descriptions, it seems that OUFBW damages gram-positive and negative bacteria by different ways. Previous your study showed that the detachment of the gram-positive cell wall from the cytoplasm. Please explain these points clearly.

<Response> Lines 336–345 (in the revised version): According to the Reviewer’s comment about the mechanisms of destruction of P. gingivalis by OUFBW, we have revised the manuscript as follows: “In this present study, OUFBW induced cytoplasmic leakage in P. gingivalis, suggesting that both the cell wall and inner membrane were disrupted. The cell wall of gram-negative bacteria comprises an outer membrane situated above a thin peptidoglycan layer [29]. The outer membrane is a lipid-protein bilayer, mainly composed of phospholipids, LPS, and membrane proteins [29]. Peptidoglycan is a heteropolymer of glycan strands crosslinked by peptides [30]. In the present study, LPS was not inactivated by OUFBW, suggesting that OUFBW does not target its active sites. This finding suggested that the outer membrane can still be disrupted by OUFBW even without LPS destruction. The inner membrane is composed of phospholipids and membrane proteins. The sulfhydryl and amino acid groups of proteins have been reported to be oxidized by ozone gas [31]. Accordingly, we considered that OUFBW damaged these spesific amino acid groups of proteins in the cell wall and inner membrane. The bactericidal mechanism of OUFBW will be more clearly understood by investigating its effects on peptidoglycan and bacterial lipid components”. Additionally, the sentence “These findings suggest that OUFBW damages Gram-negative bacteria by the same mechanism observed in Gram-positive bacteria” has been deleted because in the cited study (reference [22]), the authors did not observe any damage to the cell membrane or intracellular components of gram-positive bacteria.

Lines 212–213, 214–215, 219–220, and Figure 2 (in the revised version): TEM analysis showed that OUFBW damaged the cell walls and cell membrane and induced leakage of cellular contents. This finding indicated that OUFBW damages both the cell membrane and cell walls. The location of crack formation was considered to be between the cell wall and the cytoplasm. We added enlarged TEM images and have revised the manuscript as follows: “We observed that P. gingivalis maintained the continuous cell wall structure in the DW-treated group (Figure 2A)” in lines 212–213, “Additionally, TEM analysis showed that AUFBW created crevices, most likely between the cell wall and the cytoplasm (Figure 2B)” in lines 214–215, and “However, OUFBW damaged the cell wall and inner membrane, leading to cellular content leakage, in addition to creating crevices between the cell wall and the cytoplasm (Fig 2C)” in lines 217–218.

<Comment #6> The authors showed that the exposure condition of OUFBW for inactivation of gingipains from culture supernatant was 3.5 ppm for 1 hour. This condition seems to be harmful to human use. Were the gingipains inactivated and degraded after 30 sec exposure of OUFBW? In this regard, you should use human primary gingival epithelial cells for cytotoxicity tests because Ca922 is carcinoma cell line.

<Response> Figure S1 (in the revised version): According to the Reviewer’s suggestion, we performed an additional experiment to determine whether OUFBW inactivates gingipain within 30 s. We found that gingipain was inactivated after a 30-s exposure. Therefore, we added these data as supporting information (Figure S1).

Lines 405–409 (in the revised version): We appreciate the reviewer’s suggestion regarding the use of human primary gingival epithelial cells for a cytotoxicity assay. In this study, however, we focused on sterilization of instruments using OUFBW. We revised the manuscript as follows: “Furthermore, we considered that even if OUFBW remained on the dental instruments, it would be minimally invasive to gingival cells, as it was minimally cytotoxic against the human gingival epithelial cell line Ca9-22 (Fig 7). Currently, we do not condone using UFBWs directly on periodontal tissues; however, we would like to investigate its effect on cleaning the gingival sulcus in the future. At that time, we also plan to perform a cytotoxicity assay using primary gingival cells”.

<Comment #7> I think the use of AUFBW has potential as a disinfectant against P. gingivalis.

<Response> AUFBW may be useful as a disinfectant against P. gingivalis. However, we previously demonstrated that AUFBW could not effectively eliminate Staphylococcus aureus, as shown in Figure S1 [22], indicating that AUFBW is not effective against all bacteria.

<Comment #8> In the discussion section (page 24, line 8 from the bottom), please modify “gingivalis” to “P. gingivalis”.

<Response> Line 378 (in the revised version): We apologize for the error. This has been correctd.

Response to Reviewer 2

<Comment #1> Figure 1 and 2; In addition to DW, it is necessary to compare with a group that is not exposed to any effects (no-treated Pg).

<Response> Bacterial counts and gingipain activity measurements cannot be performed without dilution when using DW or GAM medium for P. gingivalis cultivation. To examine the difference between DW dilution and GAM medium dilution, 10 μL of bacterial culture (OD600 = 0.25) was added to 10 mL of DW or GAM medium, incubated for 30 s, and diluted using DW or GAM medium. No more than a twofold difference in viable cell counts was observed between the DW and GAM groups. Images of blood agar plates are shown (left: DW, right: GAM medium). Therefore, our results indicated that P. gingivalis does not undergo cell death in DW.

Additionally, based on the reviewer’s suggestion, we compared the DW-diluted and GAM-medium-diluted groups in Figure 2. P. ginigivalis culture supernatant (Pg sup) was diluted with DW or GAM medium and incubated for 60 min. Rgp and Kgp activities were determined using Rgp- and Kgp-specific substrates, respectively. The values of the GAM-treated group were normalized against that of the DW group (graph shown below).

As shown in the figure, the values of the GAM-treated group were reduced compared with those of the DW-treated group. This finding suggested that the measurement of fluorescence intensity was inhibited by the medium color. Therefore, dilution with DW was consider more suitable for the experiment in Figure 2.

<Comment #2>

missing number

<Comment #3> Figure 1C; There is no explanation for Figure 1c. Does this result refer to the effect of UFBW after a certain amount of time?

<Response> Lines 192–193 (in the revised version): According to the Reviewer’s comment, we have revised the manuscript as follows: “Figure 1C shows that no colonies were detected after a 3-s exposure to OUFBW”.

<Comment #4> The authors should discuss the mechanism of the bactericidal action of UFBW shown in Fig. 1AB. From Fig. 1D, can we see whether Pg bacteria burst (or do not burst) due to osmotic pressure caused by DW or UFBW?

<Response> Lines 212–213, 336–345, and Fig 2 (in the revised version): In the DW-treated group, we observed that the cell wall integrity of P. gingivalis was retained, indicating that cells were not damaged by DW. Additionally, even in the lower-magnification image of the DW-treated group (shown below), no P. gingivalis bursting was observed. Other studies have also shown that P. gingivalis is more resistant to DW-induced damage compared with the host cells (Wang et al., Science signaling, 2010). These results showed that the damaged induced in P. gingivalis by DW is not attributed to osmotic pressure. We have added enlarged TEM images in Figure 2 and have revised the manuscript as follows: “We observed that P. gingivalis maintained the continuous cell wall structure in the DW-treated group (Figure 2A)” in lines 212–213, and “In this present study, OUFBW induced cytoplasmic leakage in P. gingivalis, suggesting that both the cell wall and inner membrane were disrupted. The cell wall of gram-negative bacteria comprises an outer membrane situated above a thin peptidoglycan layer [29]. The outer membrane is a lipid-protein bilayer, mainly composed of phospholipids, LPS, and membrane proteins [29]. Peptidoglycan is a heteropolymer of glycan strands crosslinked by peptides [30]. In the present study, LPS was not inactivated by OUFBW, suggesting that OUFBW does not target its active sites. This finding suggested that the outer membrane can still be disrupted by OUFBW even without LPS destruction. The inner membrane is composed of phospholipids and membrane proteins. The sulfhydryl and amino acid groups of proteins have been reported to be oxidized by ozone gas [31]. Accordingly, we considered that OUFBW damaged these spesific amino acid groups of proteins in the cell wall and inner membrane. The bactericidal mechanism of OUFBW will be more clearly understood by investigating its effects on peptidoglycan and bacterial lipid components” in lines 336–345.

<Comment #5> In figure 1D, it is unclear whether the structure indicated by the arrowhead is an OMV. It is necessary to enlarge the image to show the structure that is continuous with the cell membrane of the Pg bacteria. If UFBW degrades OMVs and reduces their number, it may be possible to apply UFBW to OMVs isolated from the culture supernatant.

<Response> Figure 2 (in the revised version): According to the Reviewer’s suggestion, we have added an enlarged image. We observed that MVs exhibit a continuous presence in the cell membrane of P. gingivalis. Currently, our laboratory lacks the technique to isolate MVs from bacteria; thus, we plan to analyze the effect of UFBW on MVs in future work.

<Comment #6> In figure 3, Does OUFBW specifically degrade gingipain? If so, what mechanisms could be involved?

<Response>

Lines 365–366 (in the revised version): We previously de

---

## [Decision Letter · Decision Letter 1]

25 Sep 2025

Ozone ultrafine bubble water sterilizes Porphyromonas gingivalis and neutralizes its virulence factors

PONE-D-25-33041R1

Dear Dr. Terao,

We’re pleased to inform you that your manuscript has been judged scientifically suitable for publication and will be formally accepted for publication once it meets all outstanding technical requirements.

Kind regards,

Geelsu Hwang, Ph.D.

Academic Editor

PLOS ONE

Reviewers' comments:

Reviewer's Responses to Questions

**Comments to the Author**

Reviewer #1: All comments have been addressed

Reviewer #2: All comments have been addressed

2. Is the manuscript technically sound, and do the data support the conclusions?

Reviewer #1: Yes

Reviewer #2: Yes

3. Has the statistical analysis been performed appropriately and rigorously?

Reviewer #1: Yes

Reviewer #2: Yes

4. Have the authors made all data underlying the findings in their manuscript fully available?

Reviewer #1: Yes

Reviewer #2: Yes

5. Is the manuscript presented in an intelligible fashion and written in standard English?

Reviewer #1: Yes

Reviewer #2: Yes

Reviewer #1: (No Response)

Reviewer #2: The authors have adequately addressed the requests. and improved the paper's content.　The figure captions and electron microscope images have also been improved.

**Do you want your identity to be public for this peer review?** For information about this choice, including consent withdrawal, please see our Privacy Policy

Reviewer #1: No

Reviewer #2: No

---

## [Editor Report · Acceptance letter]

PONE-D-25-33041R1

PLOS ONE

Dear Dr. Terao,

I'm pleased to inform you that your manuscript has been deemed suitable for publication in PLOS ONE. Congratulations! Your manuscript is now being handed over to our production team.

Kind regards,

on behalf of

Dr. Geelsu Hwang

Academic Editor

PLOS ONE